# Investigating the Predictive Value of Thyroid Hormone Levels for Stroke Prognosis

**Aimilios Gkantzios** [1,†] **, Vaia Karapepera** [1,†] **, Dimitrios Tsiptsios** [1,*] **, Eirini Liaptsi** [1] **, Foteini Christidi** [1] **,
Elena Gkartzonika** [2] **, Stella Karatzetzou** [1] **, Christos Kokkotis** [3] **, Mihail Kyrtsopoulos** [1] **, Anna Tsiakiri** [1] **,
Paschalina Bebeletsi** [1] **, Sofia Chaidemenou** [1] **, Christos Koutsokostas** [1] **, Konstantinos Tsamakis** [4] **, Maria Baltzi** [3] **,
Dimitrios Mpalampanos** [3] **, Nikolaos Aggelousis** [3] **and Konstantinos Vadikolias** [1]

1 Neurology Department, Democritus University of Thrace, 68100 Alexandroupolis, Greece;
aimilios.gk@gmail.com (A.G.); karapeperavaya@gmail.com (V.K.); liaptsi.eirini@hotmail.com (E.L.);
christidi.f.a@gmail.com (F.C.); skaratzetzou@gmail.com (S.K.); mihoskyrtsos@gmail.com (M.K.);
anniw_3@hotmail.com (A.T.); bebeletsi@gmail.com (P.B.); haidemenousophia@gmail.com (S.C.);
chriskoutsokostas2001@gmail.com (C.K.); vadikosm@yahoo.com (K.V.)
2 School of Philosophy, University of Ioannina, 45110 Ioannina, Greece; gkartzonika@yahoo.com
3 Department of Physical Education and Sport Science, Democritus University of Thrace,
69100 Komotini, Greece; chkokkotis@gmail.com (C.K.); mmpaltzi@phyed.duth.gr (M.B.);
dimibala10@phyed.duth.gr (D.M.); nagelous@phyed.duth.gr (N.A.)
4 King's College London, Institute of Psychiatry, Psychology and Neuroscience, London SE5 8AF, UK;
ktsamakis@gmail.com
\* Correspondence: tsiptsios.dimitrios@yahoo.gr; Tel.: +30-69443-20016
† These authors contributed equally to this work.

**Abstract:** Given the expansion of life expectancy, the aging of the population, and the anticipated rise in the number of stroke survivors in Europe with severe neurological consequences in the coming decades, stroke is becoming the most prevalent cause of functional disability. Therefore, the prognosis for a stroke must be timely and precise. Two databases (MEDLINE and Scopus) were searched to identify all relevant studies published between 1 January 2005 and 31 December 2022 that investigated the relationship between thyroid hormone levels and acute stroke severity, mortality, and post-hospital prognosis. Only full-text English-language articles were included. This review includes Thirty articles that were traced and incorporated into the present review. Emerging data regarding the potential predictive value of thyroid hormone levels suggests there may be a correlation between low T3 syndrome, subclinical hypothyroidism, and poor stroke outcome, especially in certain age groups. These findings may prove useful for rehabilitation and therapy planning in clinical practice. Serum thyroid hormone concentration measurement is a non-invasive, relatively harmless, and secure screening test that may be useful for this purpose.

**Keywords:** stroke; prognosis; thyroid hormones; blood biomarkers; low-T3-syndrome; hypothyroidism; hyperthyroidism; TSH; T4; fT3; fT4

## 1. Introduction

Stroke is considered to be the principal cause of adult disability and one of the main causes of mortality worldwide. Given the heterogeneity of clinical presentation and patho-physiological mechanisms of stroke, as well as the rising incidence of stroke, there is a growing need for accurate prognostic tools to predict functional outcomes following stroke [1–5]. Modern neuroimaging techniques that facilitate precise diagnosis, combined with the use of extremely effective reperfusion therapies in the acute phase of stroke, result in lower mortality. Nevertheless, as stroke-related disability remains a significant burden, it is essential to develop new biomarkers and predictive models for optimizing post-stroke patient treatment and rehabilitation programs [6–9].

Based on the most recent Biomarkers Definitions Working Group definition, blood biomarkers must be straightforward to measure, repeatable, inexpensive, and possess both high sensitivity and specificity to be useful in therapeutic settings [10–13].

Many studies have extensively focused on the relationship between thyroid hormones, central nervous system function, and cerebral ischemia. In more detail, the thyroid hormones regulate the growth, differentiation, and maturation of brain tissue [14]. Specifically, triiodothyronine (T3) is contemplated as essential for the production and equally important maturation of new neurons, axonal myelination, and neurogenesis throughout all phases of brain development [15–17]. The structure of the cerebral parenchyma in the peri-infarct zone resembles that of the early stages of brain development, as peri-infarct regions are characterized by increased plasticity and a thriving microenvironment for remodeling. Consequently, recent data from animal model studies suggests that the same thyroid hormones may play a significant role in neuronal regeneration and plasticity and, therefore, possess neuroprotective properties [18–20]. According to the respective research, the aberrant levels of thyroid hormones may be caused by a disruption of the physiological processes involved in their metabolism and not by dysregulation of the hypothalamic-pituitary-thyroid axis. Nevertheless, the precise fundamental mechanisms remain unknown [15,21]. In animal models, T3 substitution was associated with decreased neuroinflammation and increased expression of neurotrophic factors, resulting in a more rapid motor and cognitive recovery as well as a reduced lesion volume. Additional research has demonstrated that thyroxine (T4) treatment improves neuronal and glial cell survival, facilitates brain regeneration, and increases the expression of neurotrophic factors [22]. However, it is still unknown whether these well-established neuroprotective effects of thyroid hormones in animal models can be replicated in humans.

Emerging evidence suggests that T3 has a protective effect against glutamate-induced neuronal injury, which may explain the correlation between poor stroke outcome and low blood free T3 (fT3) concentrations [23]. Other authors hypothesize that thyroid hormones may be neuroprotective in cases of ischemia via multiple mechanisms [24], including accelerating glucose transfer into brain cells [25], enhancing brain connections [26], and increasing brain vascular density [27]. In addition, multiple studies have established numerous neuroprotective effects of T3 after stroke, including a decrease in embryonic neuronal mortality and glutamate transfer to brain cells, as well as an increase in brain-derived neurotrophic factor (BDNF) [28].

Furthermore, inflammatory cytokines may play a significant role in the pathophysiological relationship between low T3 syndrome and acute ischemic stroke. Pro-inflammatory cytokines may contribute to the development of the low-T3 syndrome, as suggested by the negative correlation between IL-1, IL-6, IL-10, hsCRP, and T3 and T4. It has further been proven that an acute ischemic stroke induces a systemic inflammatory response, which is in turn followed by the production of pro-inflammatory cytokines and has been related to a decrease in T3 levels [29]. Moreover, it appears that the administration of TNF-alpha to healthy volunteers decreases T3 and thyroid-stimulating hormone (TSH) levels [30]. Additionally, evidence from animal model studies has demonstrated an interaction between pro-inflammatory cytokines and thyroid hormone metabolism [31].

Multiple studies [32,33] have also revealed that, in a model of localized ischemia, the infarction and concurrent edema are attenuated by the application of T3 by decreasing aquaporin-4 expression in peri-microvascular astrocytic end feet. In addition, T3 therapy reduces infarction and edema in a focal ischemia model, presumably by inhibiting the synthesis of inducible nitric oxide synthase, as demonstrated by in vivo studies [34,35].

Lastly, the pathophysiological mechanisms underlying decreased T3 levels in disease appear to be extraordinarily complicated and inadequately understood. Theoretically, adaptive systems that limit excessive energy consumption could be responsible for the decrease in T3. In addition, some records imply a possible association between hypothyroidism, hyperlipidemia, and endothelial dysfunction. Experts believe that low levels of fT3 may be responsible for poor collateral circulation and hypoperfusion, given these factors.

Furthermore, reduced fT3 levels may be associated with immune dysfunction, as numerous studies reveal that thyroid hormones have a significant impact on immunity [36,37].

Overall, mounting evidence suggests that low T3 levels can serve as biomarkers for predicting poor functional outcomes following a stroke, such as severe neurological impairment and mortality. These observations may be beneficial when planning rehabilitation and therapy in clinical practice. As the global burden of stroke increases in terms of the need for human and financial resources to ensure appropriate health care for the acute phase of stroke, it is crucial to develop prognostic tools for early evaluation of poor post-stroke outcomes and for guiding rehabilitation programs appropriately.

## 2. Materials and Methods

This study was guided by the Preferred Reporting Items for Systematic Reviews (PRISMA) checklist (PRISMA registration number: CRD42023442736). Our methods herein were designed a priori.

### 2.1. Search Strategy

Two researchers (AG and VK) conducted a search of two databases (MEDLINE and Scopus) to identify all relevant studies published between 1 January 2005 and 31 December 2022, using either "thyroid hormone", "hypothyroidism", "hyperthyroidism", or "euthyroid" as keywords or related terms "thyroxine", "TSH", "T3", "T4", or "triiodothyronine" as a search criterion. Moreover, the terms "stroke prognosis", "stroke outcome", or "prediction after stroke" were used as a second search criterion. In addition, there was a hand search of the retrieved articles for potentially eligible articles. Any dispute regarding the screening or selection procedure was settled by two additional researchers (DT and KV) until a consensus was reached.

### 2.2. Selection Criteria

Only original, full-text articles published in English were included. Animal studies, secondary analyses, reviews, guidelines, meeting summaries, comments, or unpublished abstracts were excluded. There were no restrictions on the study design or the sample characteristics.

### 2.3. Data Extraction

Data extraction was performed using a predefined data form created in Excel. We recorded the author, year of publication, country, number of participants, age and gender of participants, study design, scales used, and main results.

### 2.4. Data Analysis

Due to the high heterogeneity of the studies, no statistical analyses or meta-analyses were performed. Thus, the data were only analyzed descriptively.

## 3. Results

### 3.1. Database Searches

Following the database search, 331 total records were retrieved. Studies that were duplicates or irrelevant were excluded. Using our secondary search criteria, 186 full-text articles were identified as potentially qualifying for our review. After examining the complete text of the articles, thirty were deemed acceptable for inclusion (Figure 1).

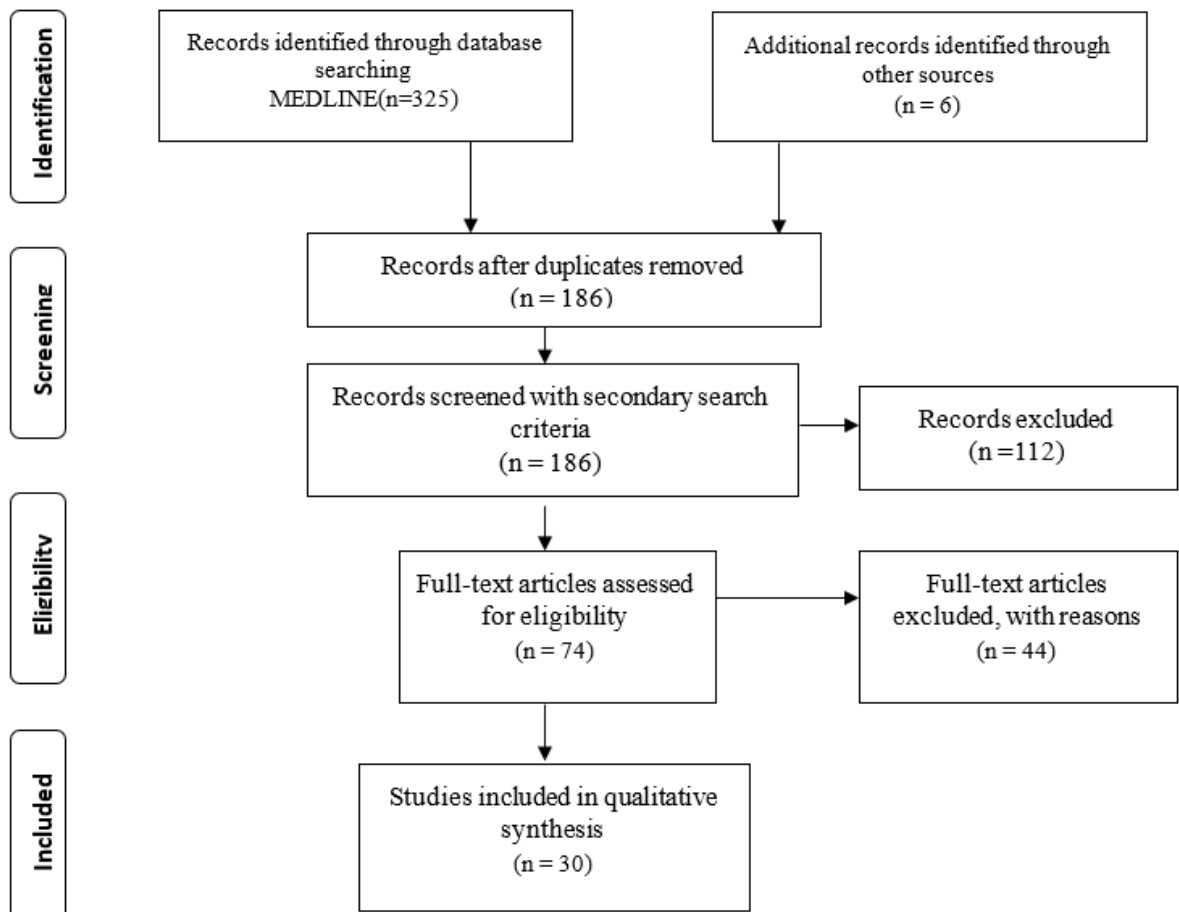

**Figure 1.** Study flow diagram (Prisma flow chart).

### 3.2. Study Characteristics

As shown in Tables 1–3, thirty publications met our inclusion criteria. The data were stratified into three categories based on the primary prognostic findings of each study in terms of clinical severity prognosis and functional outcome prognosis after stroke.

### 3.3. Study Design

In total, all studies included in this review were longitudinal. These cohorts were either prospective or retrospective.

### 3.4. Stroke Patient Groups

Across all studies, the number of stroke patients ranges from n = 46 [52] to n = 775 [48]. Four studies have a disease sample size between 1 and 100 patients; seven studies have between 101 and 200 patients; four studies have between 201 and 300 patients; four studies have between 301 and 400 patients; one study has between 401 and 500 patients; and ten studies have a disease sample size greater than 500 patients.

### 3.5. Demographic and Clinical Profiles

The mean/median patient ages ranged from 48 (IQR 45–56) years [51] to 80.1 ± 8.7 years [48]. In total, 26 studies examined patients with IS; none of the studies examined patients with HS; and 4 studies examined patients with either IS or HS.

### 3.6. Time of Blood Sampling

In 10 studies, blood sampling was conducted upon admission; in 14 studies, it was performed within the first 24 h; and in 6 studies, it was performed after the first 24 h.

**Table 1.** Studies associating thyroid hormone levels with unfavorable stroke prognosis (Basic characteristics of the studies).

| | Authors, Year of Publication | Thyroid Biomarker | Type of Study | Type of Stroke | Number of Participants/ Mean Age | Time of Blood Sampling | Scale of Stroke Severity and Prognosis/Clinical Outcome | Follow-Up Time | Cut-Off Values; (Specificity); [Sensitivity] | Main Findings |
|---|---|---|---|---|---|---|---|---|---|---|
| 1. | Alevizaki et al., 2007 [38] | T3, T4, TSH | Prospective | IS and HS | 737 patients/for low-T3: 69.9, for normal T3: 67.9 | Within 24 h of symptoms' onset | SSS, mRS | 1 and 12 months after stroke onset | Not mentioned | Low-T3 syndrome may independently predict early and late mortality as well as disability at one year in patients with acute stroke |
| 2. | Ambrosius et al., 2011 [39] | fT3, fT4, TSH | Prospective | IS | 337 patients/first tertile: 72 (61–80), second tertile: 67 (56–76), third tertile: 65 (58.8–73.5) | Within 24 h of symptoms' onset | NIHSS, mRS | • For stroke severity: on the 7th day of hospitalization. • For functional outcome: at 1 month and 360 days after stroke onset | Not mentioned | Low fT3 concentrations are associated with a poorer prognosis for stroke patients and may aid in the development of an IS outcome classification model |
| 3. | Suda et al., 2016 [40] | fT3, fT4, TSH | Retrospective | IS | 398 patients/73.3 ± 11.9 | Upon admission | NIHSS, mRS | No follow-up | For predicting poor functional outcome upon discharge: fT3 < 2.29 pg/mL; N.A. | A low fT3 value upon admission may be predictive of a poor functional outcome in patients with acute ischemic stroke |
| 4. | Wang et al., 2017 [41] | T3 | Prospective | IS | 359 patients/63.12 ± 11.3 | Upon admission | NIHSS, mRS | 1 and 3 months after stroke onset | T3: <1.34 nmol/L; (80%); [40%] | Low T3 levels can be utilized to predict the short-term prognosis of an ischemic stroke. A combined model (T3, age, and NIHSS score) can contribute substantial predictive information to the NIHSS |
| 5. | Suda et al., 2018 [42] | fT3, fT4, TSH | Retrospective | IS and HS | 702 patients/73 (64–81) | Upon admission | NIHSS, mRS | 3 months after stroke onset | fT3 < 2.00 pg/mL; N.A. | Low fT3 values at admission predict poor functional outcomes and three-month mortality in acute stroke patients |

**Table 1.** *Cont.*

| | Authors, Year of Publication | Thyroid Biomarker | Type of Study | Type of Stroke | Number of Participants/ Mean Age | Time of Blood Sampling | Scale of Stroke Severity and Prognosis/Clinical Outcome | Follow-Up Time | Cut-Off Values; (Specificity); [Sensitivity] | Main Findings |
|---|---|---|---|---|---|---|---|---|---|---|
| 6. | Zhang et al., 2019 [43] | fT3 | Retrospective | IS | 221 patients, 182 non-IS cases/patients: 66.80 ± 7.78, non-IS cases: 63.13 ± 12.51 | Upon admission | NIHSS, mRS | 3 months after stroke onset | fT3: 4.30 pmol/L; (77%); [74%] | A low fT3 value was associated with an acute IS that was particularly severe. Moreover, it predicts poor neurological outcomes three months following acute IS |
| 7. | Song et al., 2022 [44] | fT3, fT4, TSH | Retrospective | IS | 480 patients/N.A. | The following morning of the admission | NIHSS, mRS | 3 months after stroke onset | fT3 ≤ 3.69 pmol/L; (72.03%); [62.70%] | Decreased fT3 levels are a biomarker of a poor prognosis three months after an ischemic stroke |
| 8. | Zhang et al., 2010 [45] | T3, T4, TSH | Retrospective | IS | 47 patients/ 67.4 ± 12.1 | The following morning of the admission | NIHSS, mRS | 2–4 weeks after discharge from the hospital | Not mentioned | NTIS is common in patients with acute ischemic stroke, and reduced T3 levels are associated with poor outcomes. Low T3 syndrome severity may predict functional recovery in acute ischemic stroke patients |
| 9. | Huang et al., 2019 [46] | T3, fT3 | Retrospective | IS | 208 participants with HT and 208 age- and gender-matched stroke patients without HT/with HT: 68.7 ± 11.6, without HT: 68.6 ± 11.5 | Upon admission | NIHSS | No follow-up | Not mentioned | Low T3 syndrome may predict severe HT independently in AIS patients. T3 monitoring may prevent HT in patients with ischemic stroke |
| 10. | Hama et al., 2005 [47] | fT3, fT4, TSH | Prospective | IS and HS | 51 patients/ 66.7 ± 10.2 | One day after admission | FIM | No follow-up (within one week of admission and two weeks prior to discharge, all patients were evaluated for disability using the FIM) | Not mentioned | During the recovery period following a stroke, it is essential to assess the presence of NTIS by measuring free T3 and to assist patients in regaining function through intensive rehabilitation and nutritional management |

**Table 1.** *Cont.*

| | Authors, Year of Publication | Thyroid Biomarker | Type of Study | Type of Stroke | Number of Participants/ Mean Age | Time of Blood Sampling | Scale of Stroke Severity and Prognosis/Clinical Outcome | Follow-Up Time | Cut-Off Values; (Specificity); [Sensitivity] | Main Findings |
|---|---|---|---|---|---|---|---|---|---|---|
| 11. | Forti et al., 2015 [48] | fT3, fT4, TSH | Prospective | IS | 775 patients/ 80.1 ± 8.7 | On the morning after SU admission | NIHSS, mRS | No follow-up | Not mentioned | TFT at the time of hospital admission can provide independent prognostic information regarding the early outcomes of acute IS in euthyroid geriatric patients |
| 12. | Li et al., 2019 [49] | T3, T4, fT3, fT4, TSH | Retrospective | IS | 768 patients/Younger Age Group (<65): 57 (51–61), Older Age Group (≥65): 76 (70–80) | Upon admission | NIHSS, mRS | 2–4 weeks following hospital discharge | Not mentioned | Age affects total T3 level and functional prognosis in euthyroid acute ischemic stroke patients. A low total T3 concentration predicted poor functional outcomes after an ischemic stroke in adults 65 and older. None of the thyroid hormones, including total T3, independently predicted poor functional outcomes in individuals under 65 |
| 13. | Xu et al., 2016 [50] | T3, T4, fT3, fT4, TSH | Retrospective | IS | 722 patients/67 (IQR 59–76) | Upon admission | NIHSS, mRS | 2–4 weeks after discharge from the hospital | Not mentioned | Low total T3 levels predicted poor functional outcomes in ischemic stroke patients with normal thyroid-related hormone levels. Higher T4 and lower T3 values were related to greater clinical severity of stroke at admission |
| 14. | Feng et al., 2019 [51] | fT3 | Prospective | IS | 300 patients/48 (IQR 45-56) | Within 24 h after admission | NIHSS, mRS | No follow-up | Not mentioned | In early stroke, lower fT3 concentrations within normal limits are independently linked with sICAS severity and a poor prognosis |
| 15. | Liu et al., 2016 [52] | fT3, fT4, TSH | Retrospective | IS | 46 patients/ 63.6 ± 13.9 | The following morning of the admission | NIHSS, mRS | No follow-up | Not mentioned | Reduced free T3 levels are independently associated with post-IVT sICH and poor functional outcomes in patients with AIS who undergo IVT |

| | Authors, Year of Publication | Thyroid Biomarker | Type of Study | Type of Stroke | Number of Participants/ Mean Age | Time of Blood Sampling | Scale of Stroke Severity and Prognosis/Clinical Outcome | Follow-Up Time | Cut-Off Values; (Specificity); [Sensitivity] | Main Findings |
|---|---|---|---|---|---|---|---|---|---|---|
| 16. | Qiu et al., 2017 [53] | T3, T4, fT3, fT4, TSH | Prospective | IS | 159 patients/65.36 ± 10.02 | The following morning of the admission | NIHSS, mRS | 3 and 6 months after stroke onset | The cut-off value of fT3 for sICH: 3.54 pg/mL; (83%); [83%] | Low fT3 levels at admission in AIS patients receiving rtPA thrombolytic therapy were associated with sICH and inferior outcomes at 3 months |
| 17. | Ma et al., 2016 [54] | fT3, fT4, TSH | Retrospective | IS | 117 patients/58.8 ± 13.7 | The following morning of the admission | NIHSS | No follow-up | fT3 $\leq$ 4.40 pmol/L; (65.2%); [68.8%] | Low fT3 levels may play a role in the pathogenic pathway connecting inflammation to stroke severity in AIS patients |
| 18. | Suda et al., 2018 [55] | fT3, fT4, TSH | Retrospective | IS | 520 patients/ 71.9 ± 13.2 | Upon admission | NIHSS, mRS | No follow-up | The cut-off value of fT3 for PSI occurrence, fT3 < 2.29 pg/mL; N.A. | Low fT3 levels at admission are independently related to the development of PSI |
| 19. | Irimie et al., 2018 [56] | T3 | Prospective | IS | 120 patients/ 65.5 ± 9.88 | Upon admission | NIHSS, mRS, MMSE | At discharge, both functional and cognitive outcomes were assessed. No further follow-up | Cut-off value for T3 of 1.115 nmol/L to predict: <br>• stroke severity: (58.5%); [58.2%] <br>• poor cognitive prognosis: (58.9%); [66.7%] <br>• poor functional outcome: (55.4%); [70.5%] | • Higher CRP concentrations and lower T3 concentrations were correlated with stroke severity at admission <br>• CRP levels, but not T3 levels, were associated with functional outcome at discharge <br>• CRP levels were related to cognitive outcome but not T3 levels |

| Authors, Year of Publication | Thyroid Biomarker | Type of Study | Type of Stroke | Number of Participants/ Mean Age | Time of Blood Sampling | Scale of Stroke Severity and Prognosis/Clinical Outcome | Follow-Up Time | Cut-Off Values; (Specificity); [Sensitivity] | Main Findings |
|---|---|---|---|---|---|---|---|---|---|
| 20. Chen et al., 2018 [57] | T3, T4, TSH | Prospective | IS | 314 patients/ 62.97 ± 10.11 | Within 24 h of hospital admission | NIHSS, BI, MMSE | At 1 month | Not mentioned | Low T3 syndrome increased PSCI prevalence one month after an ischemic stroke |
| 21. Mao et al., 2020 [58] | T3, fT4 | Prospective | IS | 195 patients/ 69.38 ± 10.05 | Within 24 h of hospital admission | NIHSS, MoCA | For cognitive functions at 1 week, 3 months, 6 months, and 1 year | Not mentioned | A1–42 and T3 can predict the development of PSCI |
| 22. Wang et al., 2018 [59] | fT3, fT4, TSH | Prospective | IS | 634 patients/ 60.5 ± 13.1 | The second day after admission | MMSE, FSS, NIHSS | At 6 months | Not mentioned | • In patients with AIS, serum TSH levels were inversely associated with the incidence of PSF not only during the acute phase but also at follow-up • In patients with low-T3 syndrome, decreased fT3 values were related to an increased risk of PSF • TSH could predict the PSF in euthyroidism during the acute phase, but not during the follow-up phase |
| 23. Wollenweber et al., 2013 [60] | fT3, fT4, TSH | Prospective | IS | 165 patients/70 (62–78) | In the morning within 3 days after symptom onset | BI, mRS, NIHSS | At 3 months | Not mentioned | Subclinical hyperthyroidism risk outcome three months after an ischemic stroke |

**Table 1.** *Cont.*

| | Authors, Year of Publication | Thyroid Biomarker | Type of Study | Type of Stroke | Number of Participants/ Mean Age | Time of Blood Sampling | Scale of Stroke Severity and Prognosis/Clinical Outcome | Follow-Up Time | Cut-Off Values; (Specificity); [Sensitivity] | Main Findings |
|---|---|---|---|---|---|---|---|---|---|---|
| 24. | Lee et al., 2019 [61] | T3, fT4, TSH | Prospective | IS | 156 patients/ 70.3 ± 11.6 | Within 18 h of stroke onset | NIHSS | No follow-up | Not mentioned | • The group with subclinical hyperthyroidism had inferior functional outcomes, although the differences were not statistically significant • Three months after a stroke, patients with subclinical hyperthyroidism had a higher risk of poor functional outcomes and a reduced rate of successful reperfusion therapy |

Abbreviations: AIS: Acute ischemic stroke, CRP: C-reactive protein, BI: Barthel Index, FSS: Fatigue Severity Scale, FIM: functional independence measurement, fT3: free triiodothyronine, fT4: free thyroxine, GCS: Glasgow Coma Score, HS: Hemorrhagic stroke, HT: Hemorrhagic transformation, IQR: interquartile range, IS: Ischemic stroke, IVT: Intravenous thrombolysis, mBI: modified Barthel Index, MMSE: Mini-Mental State Examination, MoCA: Montreal Cognitive Assessment, mRS: modified Rankin Scale, N.A.: not applicable, NIHSS: National Institutes of Health Stroke Scale, NTIS: Nonthyroidal illness syndrome, PSCI: Postroke cognitive impairment, PSF: poststroke fatigue, PSI: Poststroke infection, SCH: Subclinical hypothyroidism, sICAS: symptomatic Intracranial Atherosclerotic Stenosis, sICH: symptomatic intracranial hemorrhage, SSS: Scandinavian Stroke Scale, SU: Stroke Unit, T3: triiodothyronine, T4: thyroxine, TFT: Thyroid function tests, TSH: thyroid-stimulating hormone.

**Table 2.** Studies associating thyroid hormone levels with favorable stroke prognosis (basic characteristics of the studies).

| | Authors, Year of Publication | Thyroid Biomarker | Type of Study | Type of Stroke | Number of Participants/ Mean Age | Time of Blood Sampling | Scale of Stroke Severity and Prognosis/Clinical Outcome | Follow-Up Time | Cut-Off Values; (Specificity); [Sensitivity] | Main Findings |
|---|---|---|---|---|---|---|---|---|---|---|
| 1. | Alevizaki et al., 2006 [62] | T3, T4, TSH | Retrospective | IS and HS | 744 patients/70.0 | Within 24 h of stroke onset | GCS, SSS, mRS | 1 and 12 months | Not mentioned | Acute stroke patients with laboratory evidence of previous hypothyroidism have a better clinical presentation and prognosis |
| 2. | Baek et al., 2010 [63] | fT4, TSH | Retrospective | IS | 31 patients with SCH and 725 patients with normal thyroid function/patients: 66.3 ± 10.8, patients with normal thyroid function: 66.2 ± 12.1 | Within two days of admission, in the morning | NIHSS, mRS | 1 and 3 months | Not mentioned | Functional results were better for AIS patients with SCH at admission |
| 3. | Akhoundi et al., 2011 [64] | T3, T4, TSH | Prospective | IS | 73 patients/66.7 | Upon admission | NIHSS, mRS, BI | 1 and 3 months | Not mentioned | Significant correlation between SCH and improved outcomes and decreased mortality after ischemic stroke |
| 4. | Delpont et al., 2016 [65] | TSH | Prospective | IS | 731 patients/ 69.4 ± 15.4 | Within the 18 h following admission | NIHSS, mRS | No follow-up | Not mentioned | A higher TSH level was independently associated with a reduced severity score at admission and a better functional outcome at discharge in AIS patients |

Abbreviations: AIS: Acute ischemic stroke, BI: Barthel Index, fT4: free thyroxine, GCS: Glasgow Coma Score, HS: Hemorrhagic stroke, IS: Ischemic stroke, mRS: modified Rankin Scale, NIHSS: National Institutes of Health Stroke Scale, SCH: Subclinical hypothyroidism, SSS: Scandinavian Stroke Scale, T3: triiodothyronine, T4: thyroxine, TSH: thyroid-stimulating hormone.

**Table 3.** Studies that conclude there is no statistically significant evidence connecting thyroid hormones with stroke outcomes (basic characteristics of the studies).

| | Authors, Year of Publication | Thyroid Biomarker | Type of Study | Type of Stroke | Number of Participants/ Mean Age | Time of Blood Sampling | Scale of Stroke Severity and Prognosis/Clinical Outcome | Follow-Up Time | Cut-Off Values; (Specificity); [Sensitivity] | Main Findings |
|---|---|---|---|---|---|---|---|---|---|---|
| 1. | Neidert et al., 2011 [66] | T3, fT4, TSH | Prospective | IS | 281 patients/68 (IQR 63–82) | 1 day after admission | NIHSS, mRS | 3 and 12 months | Not mentioned | • T3, fT4, and GH levels reflect the severity of a stroke to a lesser degree than cortisol levels • Cortisol levels are independently associated with negative outcomes after AIS |
| 2. | O'Keefe et al., 2015 [67] | fT3, fT4, TSH | Prospective | IS | 129 patients/ 67.03 ± 14.474 | At 24 ± 6 h post-symptom onset | NIHSS, mRS, mBI | 3 and 12 months | Not mentioned | Lower fT3 levels were related to worse outcomes at hospital discharge, 3 months, and 12 months following stroke, although these correlations reduced when other well-established stroke outcome predictors were considered |

Abbreviations: AIS: Acute ischemic stroke, GH: growth hormone, fT4: free thyroxine, IS: Ischemic stroke, mBI: modified Barthel Index, mRS: modified Rankin Scale, NIHSS: National Institutes of Health Stroke Scale, T3: triiodothyronine, TSH: thyroid-stimulating hormone.

*3.7. Scales of Stroke Severity, Prognosis, and Functional Outcome*

The National Institutes of Health Stroke Scale (NIHSS) and modified Rankin Scale (mRS) together were used in 17 studies. The NIHSS was the only scale in all three studies. Additionally, functional independence measurement (FIM) was the only scale in one study. In the remaining nine out of thirty studies, scales of stroke severity and clinical outcome were combined. More specifically, in two studies, the Barthel Index (BI), mRS, and NIHSS were utilized, and in one study, the modified Barthel Index (mBI), mRS, and NIHSS were modified. In one study, the Glasgow Coma Score (GCS), Scandinavian Stroke Scale (SSS), and mRS were used. In one study, the Mini-Mental State Examination (MMSE), Fatigue Severity Scale (FSS), and NIHSS were utilized. In one study, SSS and mRS were used. In one study, the NIHSS and Montreal Cognitive Assessment (MoCA) were utilized. In one study, the NIHSS, mRS, and MMSE were used. Finally, in one study, NIHSS, BI, and MMSE were utilized.

## 4. Discussion

To determine the predictive value of thyroid hormone levels as a potential biomarker for the prognosis of acute stroke, a review of the literature from the past decade was conducted. Thirty original full-text articles were identified and categorized into three categories based on their findings, which indicated a positive, negative, or no correlation between thyroid hormone levels and the prognosis of functional outcomes in stroke survivors.

*4.1. Studies Associating Thyroid Hormone Levels with Unfavorable Stroke Prognosis*

Alevizaki et al. [38] investigated the prognostic value of the low-T3 syndrome in hospitalized patients with a first acute stroke. Using the Stroke Severity Score (SSS) to evaluate stroke severity, the researchers found that individuals with low T3 were slightly but significantly older, had much more severe neurological impairment at presentation, higher mean glucose levels on admission, and a higher prevalence of cerebral oedema on CT scans. Using the Kaplan-Meier method, low-T3 patients had a lower one-year survival rate. In addition, decreased T4 levels increased 12-month mortality. Low T3 and T4 levels did not differ by type of stroke or anatomical region. According to Cox regression analysis, increased age, a history of hypercholesterolemia, a hemorrhagic stroke, a lower Scandinavian Stroke Scale score, greater hyperglycemia on admission, and lower T3 levels were independent predictors of one-year mortality. The most notable finding of this study was the independent connection between T3 levels upon admission and acute stroke patients' short- and long-term outcomes. At one year, this correlation between mortality and disability was observed. Additionally, Alevizaki et al., investigated potential connections between the aforementioned syndrome and the region of the brain most affected by the cerebrovascular event. Low T3 appeared to be associated with a poor prognosis, regardless of the location of the stroke. The only severity indicator discovered was cerebral oedema. In addition, reduced T3 levels had no effect on traditional risk factors for stroke, such as hypertension, coronary artery disease, diabetes, etc. Even when the median T3 level was used as the cutoff threshold, a correlation with enhanced survival was observed, indicating that even within the normal range, lower T3 levels are associated with a poorer prognosis. Never before has this observation been documented. Significantly, a reduced T3 concentration was associated with a more pronounced handicap at 1 year. Similarly, this association appears to be related to the severity of the initial stroke. In this direction, Ambrosius et al. [39] conducted a prospective investigation of the association between fT3 levels and the prognosis of stroke patients. The NIHSS was utilized to assess the severity of stroke at admission and on the seventh day of hospitalization. One month and 360 days after the onset of stroke, the mRS was used to evaluate the functional outcome. In both cases, poor (mRS > 2) and satisfactory (mRS ≤ 2) were the two outcome groups the patients were categorized into. The data gathered were divided into three groups in order to evaluate the significant correlations of fT3 levels on the basis of the concentration of hormone tertiles (first tertile contained the lowest levels of fT3 and third tertile contained the highest levels). Patients with fT3

levels in the first tertile were elderly and predominantly female, in contrast to those in the third tertile. In terms of patient history-related parameters, there were no significant differences between the first and third tertiles. Similar results were concluded by analyses of biochemical variables that were measured upon admission, despite the fact that patients in the first tertile of fT3 levels had higher WBC counts than those in the third tertile. In addition, the differences in glucose and urea levels between the first and third tertiles of fT3 came close to being statistically significant. The significant relationships between imaging parameters in the tertiles were also observed. When the subjects with fT3 levels in the lowest tertile were statistically compared to those in the highest tertile, compression of the ventricular system was more prevalent in those in the lowest tertile. Moreover, other important variables, namely the clinical status upon admission, the functional outcome, and mortality, varied depending on the tertile. A higher NIHSS score was associated with the first tertile of fT3 levels, while a lower score was associated with the third tertile. Patients in the lowest tertile of fT3 exhibited a greater degree of disability based on both mRS scores (30 days and 360 days) compared to those in the highest tertile. After one year of observation, patients in the first tertile of fT3 levels had a mortality rate that was significantly higher than those in the third tertile. Comparable results were revealed by the Kaplan-Mayer survival curves that depicted the relationship between mortality and tertiles. The apparent independence of the relation between overall 1-year mortality, mRS score at 30 and 360 days, and fT3 levels was confirmed by the univariate analysis, both unadjusted and adjusted for age, gender, and statistically significant variables (smoking, WBC, fT4, glucose, creatinine levels, and ventricular system compression). Concurrently, Ambrosius et al., reported that more patients with lacunar stroke had fT3 levels in the highest (third) tertile, while patients with cardioembolic etiology had the lowest fT3 levels. Ambrosius et al., drew the conclusion that low fT3 concentrations are solidly related to a poor prognosis in stroke patients, and they could thus be used to create an outcome stratification model for the said condition. Similarly, Suda et al. [40] sought to ascertain if admission levels of thyroid hormones are related to clinical characteristics and functional outcomes in patients with acute ischemic stroke (AIS). All patients included in the study were divided into quartiles according to their admission serum fT3 value and into two groups (good outcome group and poor outcome group) according to their functional outcome. Patients with poor outcomes were older, had a lower frequency of male gender, smoking, and dyslipidemia, but higher rates of atrial fibrillation and cardioembolism. In addition, the NIHSS score, glucose level, and periventricular hyperintensity (PVH) grade were significantly higher in the group with a negative outcome compared to the group with a positive outcome. Patients with poor outcomes had lower fT3 values than those with good outcomes, but no significant differences in fT4 or TSH levels were recorded between the groups. Using ROC curves, the optimal cutoff values for predicting adverse outcomes were determined for age, fT3 value, glucose level, NIHSS score, and PVH grade. In a multivariable logistic regression analysis of the aforementioned variables, age > 80 years, fT3 < 2.29 pg/mL, and NIHSS score $\geq$ 8 were independently associated with poor outcomes. After multivariate adjustment, more patients were recorded in the lowest fT3 quartile with an mRS score of 3–6 than in the highest fT3 quartile. Patients with an mRS score of 6 (death) were more prevalent in the lowest fT3 quartile than in the highest quartile after adjusting for risk factors and comorbidities; however, this difference disappeared after further adjusting for age and NIHSS score. According to the study conducted by Suda et al., reduced fT3 values were related to a poor functional outcome at the time of discharge. The said effect remained significant even after controls were conducted for risk factors and comorbidities. In addition, it is noted that the decrease in the value of fT3 equally affected the outcome in a negative manner. On the other hand, neither fT4 nor TSH levels were related to stroke severity at admission nor the functional outcome at the time of discharge.

Moreover, Wang et al. [41] investigated T3 levels with acute ischemic stroke prognosis prospectively. The NIHSS and mRS were utilized to assess clinical severity and functional outcome, respectively. A favorable outcome was characterized by an mRS score $\leq$ 2,

and an unfavorable outcome was characterized by an mRS score > 2. Patients with a poor functional outcome three months following a stroke were older, had a higher initial NIHSS, and had a lower T3 level. Poor functional outcomes were associated with a lower T3 concentration, a higher T4, fT3, fT3/fT4 ratio, a higher NIHSS score, age, glucose, BNP, C-reactive protein, white blood cells, and heart-type fatty acid binging protein at 3 months, according to univariate logistic regression analyses. Analyses of binary logistic regression revealed an association between poor outcomes and decreased T3 concentrations at admission. Furthermore, both age and NIHSS score remained significant predictors of outcome. Patients older than 68 years with a T3 concentration ≤ 1.03 nmol/L were assigned one additional NIHSS score after evaluating various combinations of possible cut-off values. They determined that a cutoff value of 1.03 nmol/L yielded an AUC of 0.70 with a sensitivity of 83% and a specificity of 60%. Their laboratory's cutoff value for low T3 concentration (1.34 nmol/L) exhibited a sensitivity of 40% and a specificity of 80%. Based on the findings of their study, they created a model that incorporated triiodothyronine, age, and the NIHSS score. They designated this model as NIHSS-TA. With extended follow-up, they discovered that NIHSS-TA improved the accuracy of prognosis prediction. To sum up the findings of their study, the authors reached the conclusion that low T3 levels can be utilized in the prediction of the short-term outcome following an ischemic stroke. In addition, a combined model (T3, age, and NIHSS score) can add significant predictive information to the NIHSS clinical score. In addition, Suda et al.'s [42] aim was to ascertain whether the admission serum concentrations of TSH, fT3, and fT4 are associated with 3-month outcomes in patients with acute stroke. A multivariate logistic regression analysis revealed that a low fT3, older age, a high preadmission mRS score, and a high NIHSS score independently predicted a poor functional outcome. In addition, low fT3, old age, diabetes, the preadmission mRS score, the NIHSS score, and hemorrhagic stroke were independently associated with mortality. After controlling for age, gender, preadmission mRS score, prior stroke, stroke severity on admission, and several comorbidities, this effect remained significant. Likewise, Zhang et al. [43] retrospectively collected data on 221 cases of AIS and 182 non-AIS cases from their hospital. They used multivariable models that predicted three-month functional outcomes from admission fT3 concentrations. ROC curve calculations determined the appropriate fT3 cutoff value for stroke severity and subtypes as well as for neurological outcome. The correlation between fT3 and stroke risk variables revealed that AIS patients with NIHSS on admission greater than 3 and 3-month mRS greater than 2 were more likely to have low fT3. TIA history, hypertension, diabetes, hyperlipidemia, hyperhomocysteinemia, smoking, alcohol consumption, atrial fibrillation, stroke history, and cerebral small vessel-related lesions had no association with the fT3 level. fT3's ROC curve analysis showed a threshold value of 4.30 pmol/L with 74% sensitivity and 77% specificity for predicting stroke severity on admission. fT3 levels also differed amongst stroke subtypes, with the large artery atherosclerosis (LAA) and cardioembolism (CE) subtypes having moderately lower levels than the small vessel occlusion (SVO) subtype. More specifically, the fT3 values of patients with LAA and CE were lower than those of patients with SVO, but there were no significant distinctions between LAA and CE. The fT3 threshold for the LAA group was 4.18 pmol/L (sensitivity: 71%; specificity: 73%). 4.31 pmol/L (sensitivity: 52.0%; specificity: 83.0%) was the concentration of the CE group (sensitivity: 52.0%; specificity: 83.0%). The SVO group's threshold was 4.68 pmol/L (sensitivity: 42%; specificity: 80%). In univariate logistic regression studies, a negative clinical outcome was associated with a reduced fT3 concentration, a higher NIHSS score, and an older age. According to binary logistic regression, poor outcome and low admission fT3 were also connected. Age and NIHSS entry are also associated with clinical outcomes. Zhang et al., concluded, on the one hand, that fT3 was an independent and significant predictor of 3-month neurological outcome and, on the other hand, that a decrease in fT3 value was correlated with the severity of AIS. The ROC analysis also demonstrated that AIS patients with a fT3 concentration of 4.38 pmol/L possessed a robust predictor of the neurological outcome (sensitivity 78%; specificity 73%). Song et al. [44] examined

how admission thyroid hormone levels affected functional outcomes three months after acute ischemic stroke in 480 patients in their single-center, retrospective, observational analysis. Poorer outcomes were linked to greater fT4 but lower TSH and fT3. Female gender, atrial fibrillation, coronary heart disease or heart failure, recanalized treatment, older patients, lower hemoglobin, albumin, eGFR, and white blood cell counts, and higher NIHSS scores are all related to poor outcomes. In terms of the association between gender and severity of outcome after stroke, previous research indicates that low fT4 values are associated with a more severe clinical profile and a poorer functional outcome in men than in women. These results contradict the findings of the included study in this review. The preceding highlights the need for additional research into the association between gender and thyroid hormones in stroke patients [68,69]. After a multivariate analysis, whose main goal was to identify independent risk variables for poor outcomes at 3 months, it was discovered that lower fT3 levels, age, NIHSS score, and no recanalization therapy were linked with worse functional outcomes. TSH and fT4 levels were not significant after the respective adjustments for other parameters. Song et al., concluded that decreasing fT3 levels adversely impacted functional outcomes. Additionally, Zhang et al. [45] examined thyroid hormone status, clinical severity as measured by the NIHSS, and outcome in acute ischemic stroke patients, as well as the relationship between pituitary axis abnormality and anterior or posterior circulation involvement. All patients were separated into low-T3 and normal-T3 groups based on their thyroid values on arrival. On admission, NIHSS scores were classified as mild, moderate, or severe. An improvement in NIHSS and mRS is considered a good outcome. Approximately two to four weeks after hospital discharge, the first clinic appointment was scheduled. The study discovered that individuals with low T3 levels had worse neurological impairment and higher mean glucose levels upon admission. The group with low T3 had considerably higher NIHSS scores than the one with normal T3. The admission distribution pattern of NIHSS scores showcased that the majority of low-T3 patients were moderate-to-severe, while the majority of normal-T3 patients were mild. TSH levels were comparable between groups with reduced and normal T3. There was no difference between low-T3 and normal-T3 groups for strokes that affected the anterior or posterior circulation. An adverse correlation between T3 levels and NIHSS scores was noted for all patients. This suggests that neurological damage is worse following an acute stroke with decreased T3 levels. During the initial stroke outpatient clinic follow-up, fewer patients with low T3 presented optimal neurological function on the NIHSS (27.6% versus 66.7%) and mRS (31% versus 55.5%) scales. Low T3 levels are related to deteriorating neurological outcomes. The severity of low T3 syndrome in AIS patients may indicate functional recovery. This study suggests that thyroid hormone monitoring following a stroke should be strongly considered.

From an alternative perspective, Huang et al. [46] hypothesized that AIS patients with low T3 syndrome are susceptible to developing hemorrhagic transformation (HT). Researchers evaluated the effects of low T3 syndrome on HT in AIS patients to address this issue. HT patients had a mean age of 68.7 years, while those without HT had a mean age of 68.6 years. 60.1% of HT patients had asHT, while 39.9% had sHT. 24.0% of patients had HI-1, 30.8% had HI-2, 23.1% had parenchymal hematoma type 1, and 22.2% had type 2. In a comparison attempted among the demographic, clinical, and laboratory characteristics of patients with and/or without HT, those with HT were more likely to receive anticoagulant or atrial fibrillation medication and less likely to receive antiplatelet or lipid-lowering therapy. Their leukocyte, fibrinogen, total cholesterol, glucose, and NIHSS scores at the start of the study were higher. Additionally, smokers and alcoholics were less prevalent among HT patients. Total and free T3 levels at baseline were substantially lower in HT patients compared to non-HT patients. Low T3 syndrome was more prevalent in HT patients admitted with an ischemic stroke. HI had significantly higher total and free T3 levels at baseline than PH. Low T3 syndrome was less prevalent in patients with HI than in patients with PH. The outcomes of the patients with sHT and asHT were comparable. In univariate studies, low T3 syndrome at baseline was always significantly associated

with radiological and clinical HT and PH risks over HI. Each dependent variable was represented by identical covariates in adjusted multivariable models. Low T3 syndrome predicted HT independently after adjusting for age and gender. After adjusting for variables associated with HT risk, the OR of HT for low T3 syndrome showed no changes. After controlling for other characteristics that notably differed between HT patients and non-HT patients in the univariate analysis, the relationship remained. Low T3 syndrome and sHT3 showcased comparable symptoms. Low T3 was a significant and independent predictor of HT classification (PH or HI) after controlling for confounding variables. Low T3 syndrome did not affect HI. According to the authors, this was the first study to evaluate low T3 syndrome and HT in AIS patients. Low T3 syndrome was independently associated with HT, especially PH and sHT, after controlling for all variables.

Sick euthyroid syndrome, or nonthyroidal illness syndrome (NTIS), seems to occur when protein-energy malnutrition (PEM) and other maladies decrease T3 without increasing TSH. Taking this into account, Hama et al. [47] sought to determine how PCM and NTIS affected the functional dependency of stroke patients. Blood albumin levels and BMI, which are sensitive indicators of PCM, were used to measure malnutrition. According to the severity of NTIS, the patients were divided into two subgroups (mild and severe NTIS). More specifically, mild NTIS was characterized by a decrease in free T3 and normal free T4 and TSH. Severe NTIS was characterized by a decrease in free T3 and T4 without elevating TSH. Patients with strokes were evaluated using the functional independence measurement (FIM). Based on their initial FIM score, this scale divided patients into two groups: low-FIM (70 or less) and high-FIM (71 or more). Age, gender, stroke type, and medical history were elements that proved able to be compared between the said groups. 57% of patients exhibited hypoalbuminemia, 22% were underweight, and 82% were diagnosed with moderate NTIS. None had severe NTIS. Patients with a high FIM had elevated levels of albumin and BMI. Free T3 had a significant positive correlation with serum albumin and body mass index. The good outcome group had substantially higher levels of free T3, but not albumin or BMI. PCM is found to be associated with non-thyroidal disease in many stroke survivors with functional dependence. NTIS contributes to impaired functional recovery in stroke patients. Therefore, according to this study, it is essential to test for NTIS and provide extensive rehabilitation and nutritional care following a stroke.

Considering the age of stroke patients and 65 years as a plausible threshold for distinguishing thyroid hormone behavior, Forti et al. [48] evaluated whether TSH, fT4, and fT3 measured upon admission are associated with early outcomes of acute IS in euthyroid patients above 65 years old. Composite outcomes included poor functional outcomes and adverse discharge settings. Discharge mRS $\geq 3$ was characterized as disability if prestroke mRS was between 0 and 2, or discharge mRS greater than prestroke mRS if prestroke mRS was greater than 3. About one-third of subjects had fT3 levels that fell below the normative range. Age was associated with increased TSH and fT4 levels but decreased fT3 levels. The primary cause of IS was cardioembolism. Lower TSH, greater fT4, and lower fT3 were associated with higher NIHSS admission scores. Patients with poor functional outcomes had lower TSH and fT3, while those released to unfavorable settings had greater fT4. After adjusting for demographic, AF, and IS confounders, only the association between reduced fT3 and an unfavorable discharge environment remained statistically notable. Adjusted logistic regression models for TSH demonstrated a quadratic relationship with a poor discharge functional outcome but not with an adverse discharge environment. In addition to interacting with age, the linear and quadratic TSH factors predicted inferior functional outcomes. At age 65, the probability of a poor functional outcome decreased when TSH was increasing; it reached a minimum near 3.00 mUI/L and then increased. However, between the ages of 70 and 75, the curve became straight, and, beginning at age 80, the curve assumed an inverted U-shape: outcome risk rose with increasing TSH, reached its peak at TSH values that progressively shifted upward with age (from 1.70 mUI/L at 80 to approximately 2.30 mUI/L at 90), and then began to decline. In the adjusted logistic regression models for peripheral thyroid hormones, fT4 did not

affect the study outcomes. The adjusted logistic regression models for fT3 demonstrated a statistically significant negative linear relationship with an unfavorable discharge setting (risk decreased by 60% per unit increase in fT3). This study found that a single thyroid function test (TFT) on SU admission can predict the outcome of acute IS in euthyroid older individuals. These findings may aid in the risk stratification of geriatric IS patients and the development of neuroprotective medications. Similarly, Li et al. [49] divided 768 patients with acute ischemic stroke into two age groups: a younger group (age < 65 years) and an elder group (age > 65 years) to evaluate how age affects thyroid hormone prognosis after an AIS. According to univariate analysis, younger patients with worse functional outcomes had lower total and free T3 concentrations, higher NIHSS scores, and atrial fibrillation. In addition to the aforementioned characteristics of the younger group, older age, female gender, greater free T4, and lower TSH levels are linked to poor function in the older cohort in univariate analysis. As determined by multivariate logistic regression analysis, only NIHSS scores were independently related to poor functional outcomes in the youngest group. In this group, neither total nor free T3 independently predicted poor function. As far as the older population is concerned, reduced total T3 levels were independently related to poor functional outcomes. Li et al., highlighted the fact that total T3 level and functional outcome vary with age in euthyroid acute ischemic stroke patients. Lower total T3 concentration was an independent predictor of poor functional outcome following ischemic stroke in patients aged 65 and older. Nonetheless, none of the thyroid hormones independently predicted poor functional outcomes in individuals under 65.

Xu et al. [50] investigated the prognostic value of thyroid hormones within normal ranges in patients with AIS. The NIHSS was used to assess the severity of stroke at admission, and the distribution of NIHSS scores at admission was classified as follows: mild (NIHSS < 8), moderate (NIHSS 8–14), and severe ($\geq$14). At the initial follow-up clinic visits, which typically occurred 2 to 4 weeks following the hospital discharge, the mRS was used in order to evaluate functional outcomes (unsatisfactory outcomes: mRS > 2, satisfactory outcomes $\leq$ 2). The researchers discovered negative correlations between NIHSS scores and total and fT3 levels, as well as between CRP levels and total and fT3 levels. Positive correlations were observed between NIHSS scores and concentrations of total and fT4, but not between TSH or CRP and fT4 levels. In addition, CRP levels and ratios of fT3/fT4 hormones were found to have a negative correlation. In univariate analysis, the existence of atrial fibrillation as the index event was related to a poor functional outcome. Moreover, the NIHSS, total T4, fT4, and CRP levels at admission were notably higher in patients with a poor functional outcome, whereas fT3 and total T3 levels were notably lower. Patients with positive functional outcomes exhibited significantly higher fT3/fT4 ratios than those with negative functional outcomes. The said finding propounds a decrease in peripheral T4 to T3 conversion in suboptimal functional outcome patients. In multiple logistic regression analyses, reduced total T3 concentrations remained independently associated with poor functional outcomes. NIHSS scores were the only other variable associated independently with poor functional outcomes, in addition to total T3. To sum up, as far as the findings were concerned, it was concluded that a low total T3 concentration was an independent predictor of poor functional outcome in ischemic stroke patients with normal levels of thyroid-related hormones. Greater stroke severity at admission was linked to lower T3 (total and free) and higher T4 (total and free) levels. Similarly, Feng et al. [51] evaluated the connection between thyroid hormones within normal limits, WBC count, initial stroke severity, and early functional outcomes 14 days after admission in AIS patients with intracranial atherosclerotic stenosis (ICAS). Hypertension, WBC, Fib, and fT3 predicted initial severity in univariate research. Logistic regression analysis highlighted the fact that hypertension, lower fT3 within normal ranges, and greater WBC concentrations could independently predict severe stroke. In terms of functional outcome, WBC, plasma glucose, Fib, fT4, and NIHSS score on admission were significantly higher in patients with poor outcomes in comparison to patients with favorable outcomes, whereas diastolic blood pressure (DBP), albumin/globulin (A/G), and fT3 were significantly lower. The analy-

sis of logistic regression revealed that abnormally low fT3, elevated fT4, plasma glucose, and white blood cell count remained independently associated with adverse outcomes. Feng et al., observed in their study that comparatively low fT3 concentrations within normal ranges are independently associated with initial stroke severity on admission and worse functional outcomes 14 days after admission. The underlying mechanisms were unidentified. In addition, they investigated the association between WBC count and patients with symptomatic ICAS (sICAS) and reached the conclusion that a high WBC count was linked to both early poor functional outcomes and initial stroke severity. Lower fT3 concentrations and higher WBC levels independently predicted stroke outcomes, and bivariate correlation analyses showed a negative connection.

Concerning the potential relationship between thyroid hormone levels and the prognosis of patients with symptomatic intracranial hemorrhage after intravenous thrombolysis in patients with ischemic stroke, Liu et al. [52] conducted the first study that investigated the relationship between thyroid hormone levels and the prognosis of patients with symptomatic intracranial hemorrhage (sICH) after intravenous thrombolysis in ischemic stroke patients. For their research, they divided fT3 into the following quintiles: Q1: < 3.44 pmol/L, Q2: 3.44–3.71 pmol/L, Q3: 3.71–4.43 pmol/L, and Q4: > 4.43 pmol/L. The first quintile had higher admission NIHSS scores, urea, creatinine, and post-IVT sICH. Eleven patients (91.7%) in the first quintile of fT3 levels had unsatisfactory outcomes at discharge, as did 5 (42.7%) in the second, 4 (36.4%) in the third, and 2 (18.2%) in the fourth. At discharge, one patient in the first quintile of fT3 levels perished. After univariate and multivariate analyses, fT3 levels were linked to post-IVT sICH or poor functional outcomes when discharged. Low thyroid function was associated with higher urea and creatinine levels, suggesting decreased renal function. Moreover, they found that post-IVT sICH and poor functional outcome at discharge in AIS patients are independently linked with lower fT3 levels. In the same direction, Qiu et al. [53] investigated thyroid hormones and outcomes in AIS patients treated with intravenous rtPA to find a new predictor for sICH and poor outcomes following thrombolysis. The NIHSS assessed neurologic impairment. The outcomes at 3 or 6 months were evaluated using the mRS; they defined mRS 0–2 as satisfactory and mRS greater than 2 as poor. First, they determined the unadjusted values for fT3, fT4, tT3, tT4, TSH, and fT3/fT4 for each outcome category. Patients with sICH had decreased fT3 and fT3/fT4 levels. Patients with poor outcomes had decreased fT3 and fT3/fT4 ratios. fT4 levels had no significant impact on any of the outcome categories. The upper quartiles of fT3, fT4, and fT3/fT4 showcased a lower incidence of sICH and favorable outcomes at 3- and 6-month follow-up. fT4 had no effect on sICH or outcomes. During the course of their inquiry, they assessed outcomes and fT3 using multivariable logistic regression to adjust for other risk factors. The NIHSS score was an independent risk factor for sICH following thrombolytic therapy, but fT3 was a substantial protective factor. The same outcomes were observed in the 3-month follow-up of the patients with unsatisfactory outcomes. Moreover, NIHSS score and baseline blood glucose were independent risk factors for poor 6-month outcomes, but fT3 was not. Finally, fT4, tT3, tT4, TSH, and fT3/fT4 were not independently related to any of the outcomes. Qiu et al., found that fT3 was strongly linked with sICH and 3-month outcomes in IV rtPA-treated AIS patients. After controlling for confounders, lower fT3 was an independent risk factor for sICH and poor functional outcomes. AIS patients with fT3 levels of 3.54 pg/mL or less had a 3.16-fold increased risk of developing sICH after receiving IV rtPA. Finally, neither tT3, fT4, tT4, nor TSH independently predicted sICH or functional results.

Regarding the possible relation between the inflammatory process observed during the acute phase of stroke, the levels of thyroid hormones, and, by extension, the prognosis after stroke, Ma et al. [54] examined how lowered fT3 and inflammatory markers such as CRP, FIB, albumin, ESR, and WBC affect stroke severity in AIS patients. Patients were classified as mild (NIHSS ≤ 5) or moderate (NIHSS between 6 and 15). The mild group had higher fT3 concentrations than the moderate group, but fT4 and TSH concentrations were similar. Moreover, fT3 concentration was inversely connected with NIHSS score, CRP,

ESR, and fibrinogen and positively correlated with albumin. The ROC curve revealed that fT3 ≤ 4.40 pmol/L was the best predictor of stroke severity, with a sensitivity of 68.8% and a specificity of 65.5%, and can be used as the optimal cutoff value. Multiple regression corroborated the relationships between fT3 and CRP, albumin, and the NIHSS score. To reduce confounding variables for stroke severity, a binary logistic regression was used. This data implies that fT3 levels in AIS can discriminate stroke severity. In regression analysis, fT3/NIHSS was linked with albumin, CRP, and TG concentrations. These results indicate a pathogenic mechanism coupling inflammation to stroke severity. Moreover, commonly following a stroke is an infection. Post-stroke infections are known to exacerbate stroke outcomes and increase mortality. Infection risk is increased by stroke, dysphagia, and aging. Immune suppression caused by a stroke, which may be a result of sympathetic nervous system activity, increases the risk of infection. In light of this, Suda et al. [55] examined the relationship between admission thyroid hormone levels and PSIs in acute ischemic stroke patients. They enrolled patients retrospectively and assessed stroke severity using the NIHSS score at admission. PSIs included pneumonia, UTIs, and other infections. The study involved 520 patients, 107 of whom developed PSI. The most prevalent infection was pneumonia (60.7%), followed by urinary tract infections (17.8%). Patients with PSIs were older and had significantly higher NIHSS scores at admission, blood glucose levels, mRS scores at admission, mortality rates, and mRS scores when discharged. PSI patients had lower fT3 levels than non-PSI patients, although fT4 and TSH levels were similar. A multivariable logistic regression analysis incorporating age, fT3 level, blood glucose level, body mass index, NIHSS score at admission, and mRS score at admission revealed that PSI was independently associated with a fT3 level of 2.29 pg/mL or less. The PSI incidence rates based on the fT3 cutoff value were 13.1% for fT3 levels above 2.29 pg/mL and 39.7% for fT3 levels at or below 2.29 pg/mL. From this investigation, Suda et al., found that 20% of patients had PSI and that it was associated with poor functional results. After controlling for several risk variables and comorbidities, lower fT3 levels were related to a higher incidence of PSIs. In contrast, PSI rates were not linked with serum fT4 or TSH levels. Additionally, Irimie et al. [56] evaluated CRP and T3 as independent predictors of poor cognitive and functional outcomes in patients with acute ischemic stroke upon hospital discharge. In their study, 120 individuals with acute ischemic stroke participated. The mRS and the MMSE were used to evaluate disability and dependence, respectively. Specifically, they discovered that a poor functional outcome, as determined by the mRS, was positively correlated with serum CRP levels and negatively correlated with T3 concentrations at admission. Concerning cognitive impairment, there was a negative relationship between MMSE score and CRP levels and a positive relationship between MMSE score and T3 levels. In binary logistic regression, higher admission CRP concentrations predicted poor cognitive outcomes at discharge but not functional outcomes. Lower admission fT3 concentrations did not increase discharge cognitive or functional impairment. Using ROC curve analysis, they determined a serum CRP cutoff level of 3.035 mg/dL to predict a severe stroke with a sensitivity of 79.7% and a specificity of 75.6%, a poor cognitive prognosis using the MMSE with a sensitivity of 75.6% and a specificity of 83.3%, and a poor functional prognosis using the mRS with a sensitivity of 83%. Additionally, it has been determined that a T3 cutoff value of 1.115 nmol/L can predict stroke severity with a sensitivity of 58.2% and a specificity of 58.5%, a poor cognitive prognosis with a sensitivity of 66.7% and a specificity of 58.9%, and a poor functional outcome with a sensitivity of 70.5% and a specificity of 55.4%. Finally, a ratio of CRP and T3 levels indicated a 53.5% poor functional outcome and an 80.4% poor cognitive outcome in stroke patients at the time of discharge. These findings demonstrate that a high CRP level and lower T3 concentrations can be utilized as a prognostic sign for ischemic stroke and a predictor of a severe course.

Poststroke cognitive impairment (PSCI) is a common stroke consequence that worsens functional results, quality of life, and recurrence risk. Chen et al. [57] investigated the potential link between thyroid hormones (specifically low T3 syndrome) and PSCI in the acute phase of ischemic stroke and at one-month follow-up. 314 consecutive ischemic stroke

patients were observed for one month in the study. Cognitive function was defined as an MMSE score of less than 27 one month following an acute ischemic stroke, and thyroid hormones were evaluated within 24 h of admission. Compared to non-PSCI patients, PSCI patients had decreased T3 levels. In addition, 56.6% of PSCI patients had low T3 syndrome, compared to 33.3% of non-PSCI patients. Moreover, PSCI was also considerably higher in low-T3 syndrome individuals. After correcting for confounders, low T3 syndrome independently predicted PSCI prevalence. Finally, after controlling for covariates, low T3 syndrome was independently related to PSCI, and older age and poorer education were risk factors for cognitive impairment in their logistic model. Moreover, Mao et al. [58] investigated the association between thyroid hormones and serum levels of beta-amyloid protein 1–42 in the acute phase of stroke and post-stroke cognitive impairment (PSCI), as well as its predicted value. After several measures, they highlighted the fact that the levels of Aβ 1–42 and T3 in the PSCI group initially decreased and then increased over time, reaching a low level at 3 months, stabilizing at 6 months, and then progressively rising. In contrast, the levels of Aβ 1–42 and T3 in the non-PSCI group increased continuously, with a steady but progressive pace. The levels of Aβ 1–42 and T3 peaked within one week of onset and then decreased marginally, exhibiting a consistent trend. fT4 levels in both groups increased continuously over time, but there was no significant difference between them overall. During the follow-up, Aβ 1–42, T3, and fT4 were positively linked to disease progression. After adjusting for age, gender, education level, body mass index, smoking, alcohol, disease history, NIHSS, LDL, and TG, the risk of PSCI increased with decreasing Aβ 1–42 and T3 levels in patients older than 70 who were female and illiterate. After a one-year follow-up of stroke patients, Mao et al., discovered that due to the time course of Aβ 1–42 and T3, the management of patients three to six months after a stroke should be enhanced so as to detect and prevent PSCI as quickly as possible. In addition, repeated measurements highlighted the fact that the overall content of Aβ 1–42 and T3 was lower in the PSCI group than in the non-PSCI group, thus indicating that the changes in Aβ 1–42 and T3 following a stroke could be related to the progression of the disease and that both can dynamically monitor and evaluate the risk of disease over time. Nevertheless, it should be mentioned that the analysis must be expanded due to the extent of the influence of the subsequent changes in indicators. A Spearman analysis and a regression analysis indicated that Aβ 1–42 and T3 were significant factors that influenced the way the disease occurred, regardless of whether or not confounding factors were taken into consideration. This said, it is indicated that Aβ 1–42 and T3 have a promising future in assessing the way PSCI occurs and progresses.

Poststroke fatigue (PSF) is a subtype of pathologic fatigue, a debilitating symptom that affects 23 to 85 percent of stroke patients. It may negatively impact stroke survivors' rehabilitation and long-term outcomes. Patients, families, and even health care professionals frequently overlook or underrecognize PSF. Therefore, Wang et al. [59] investigated whether the thyroid function profile and serum levels could predict PSF in a prospective stroke cohort. The study utilized the 9-item FSS, which is the most commonly employed instrument for assessing PSF, as a higher score indicates a greater degree of fatigue. In their analyses, fatigue was identified in patients with a mean FSS score of 4 or higher. They applied the FSS at the acute phase of stroke and reevaluated with the same scale six months later. Patients were divided into four distinct groups: euthyroidism, subclinical hypothyroidism, subclinical hyperthyroidism, and low-T3 syndrome. The proportions of the four patient categories were, respectively, 71.9%, 9.6%, 6.0%, and 12.5%. Following a stroke, the prevalence of PSF in the acute phase was 41.5%, and at the 6-month follow-up, it was 35.3%. In the assessment of acute phase fatigue, higher TSH levels were associated with a lower risk of PSF. TSH levels also differed significantly between the euthyroidism subgroups. Adjusting numerous variables revealed that the TSH level was a protective biomarker for PSF in both the entire patient population and the euthyroidism subgroup. A multiple linear regression analysis was conducted to further investigate the correlation between thyroid function and PSF. TSH was negatively correlated with PSF at the acute

phase of stroke in the euthyroid group, whereas fT4 was positively correlated with PSF. In the subclinical hypothyroidism group, TSH and fT4 levels were also linked to PSF; in the low-T3 syndrome group, TSH and fT3 levels were independently associated with PSF in multivariate analysis. In both the euthyroid and low-T3 syndrome groups, the risk of PSF increased with rising fT4 levels at the six-month follow-up. TSH was a protective predictor for PSF, according to the findings of this study. The incidence of PSF decreased by 70% for each TSH increment in the full range of thyroid function and by 60% for each TSH level in the full range of thyroid function. There was no correlation between thyroid function and PSF in the normal range of thyroid function. The risk of PSF was reduced by 30% in patients six months after the stroke whose TSH levels were elevated. Finally, according to their study, fT3 was a significant predictor of the risk of PSF in patients with low-T3 syndrome.

Subclinical hyperthyroidism increases the risk of carotid plaques and stroke, as well as several other cardiovascular illnesses and ailments. Considering the preceding, Wollenweber et al. [60] decided to take a closer look at the effect of subclinical hyperthyroidism on functional outcome following a stroke. In their prospective observational study, 165 stroke patients were evaluated and divided into three groups: subclinical hyperthyroidism, subclinical hypothyroidism, and euthyroid state. Three months following a stroke, individuals were monitored. Functional impairment (mRS) and dependency (Barthel Index) were the main outcomes. They discovered that subclinical hyperthyroidism was associated with a lower body mass index and blood lipid levels. Additionally, subclinical hyperthyroidism and hypothyroidism increased the risk of atrial fibrillation. In statistical models, subclinical hyperthyroidism was linked to a significantly increased risk of a negative functional outcome. Nonetheless, subclinical hypothyroidism was associated with a decreased risk of a poor functional outcome, despite the fact that the association between them was not statistically significant. Three months after an ischemic stroke, subclinical hyperthyroidism independently predicts a poor functional prognosis, according to Wollenweber et al. Atrial fibrillation (AF) and cardioembolic stroke, which have a worse prognosis than the other subtypes of stroke, are both increased by subclinical hyperthyroidism. Similarly, in an effort to identify a new predictor of poor outcomes following reperfusion, Lee et al. [61] examined the correlations between subclinical thyroid dysfunction (SCTD) and the outcomes of patients with acute ischemic stroke who were treated with reperfusion therapy (intravenous thrombolysis (IVT), intraarterial thrombectomy (IAT), or a combination of IVT). Due to the diurnal variation in TSH, late morning and daytime (09:00 a.m. to 15:00 p.m.) blood samples were obtained within 18 h of the onset of stroke. Three groups of patients were formed: subclinical hyperthyroidic (SCHyper, TSH < 0.35 U/mL), subclinical hypothyroidic (SCHypo, TSH > 4.94 U/mL), or euthyroid (0.35 µU/mL $\leq$ TSH $\leq$ 4.94 µU/mL) with normal fT4 and total T3 levels. Hundred and fifty-six acute ischemic stroke patients underwent reperfusion therapy from 1360 consecutively enrolled stroke patients. In general, the demographics and stroke characteristics of the three SCTD groups were comparable. The probability of poor functional outcomes at 3 months was higher for the SCHyper than the euthyroid and SCHypo groups, according to multivariate analyses. Additionally, three months following IV tPA, a higher risk of poor functional outcomes was noted for SCHyper patients than for the euthyroid and SCHypo ones. Finally, a further finding of the researchers was that SCHyper patients with major arterial steno-occlusion before reperfusion exhibited a worse success rate than euthyroid and SCHypo patients.

*4.2. Studies Associating Thyroid Hormone Levels with Favorable Stroke Prognosis*

Alevizaki et al. [62] correlated the thyroid status of hospitalized patients with acute stroke severity and clinical outcome. Within twenty-four hours of exhibiting symptoms, 744 people underwent thyroid function testing. Their analysis included a number of stroke risk factors as well as the Glasgow Coma Scale (GCS) and Scandinavian Stroke Scale (SSS) in order to determine stroke severity upon admission. TSH levels within the range of 0.4 to 3.2 U/mL are considered normal. Thirteen people had elevated TSH ($\geq$10 µU/mL;

range 10–42 µU/mL), 51 had marginally elevated TSH (3.3–9.9 µU/mL), and 680 had TSH <3.3 µU/mL. On admission, hypothyroid patients had substantially lower mean T4 and glucose levels. Moreover, patients with hypothyroidism had substantially improved GCS and SSS scores. The incidence of the two most common forms of stroke, ischemic and hemorrhagic, was identical. Individuals with hypothyroidism had a greater incidence of TIAs. There was a 12-month functional status follow-up for 637 individuals. Although this difference was not statistically significant, hypothyroid patients had a higher favorable outcome score. Alevizaki et al., discovered a substantial correlation between hypothyroidism and a favorable outcome in patients with acute stroke. This protective effect has not been noted or examined before and may be the result of decreased adrenergic tone, a slowing metabolism, or other unidentified factors in the patients under examination. In addition, previous transient ischaemic attacks may have a protective effect. Baek et al. [63] additionally proceeded with an evaluation of the relationship between SCH and the functional outcomes of acute ischemic stroke. Thyroid function has been assessed by measuring TSH and fT4 levels, and clinical evaluation includes NIHSS and mRS scores on admission and 30 days after stroke. On day 90 following a stroke, mRS scores were re-evaluated for functional outcomes. Two criteria evaluated the effects of a stroke at 30 and 90 days. In the first outcome model, favorable outcomes [I] were defined by mRS scores (≤1). T he favorable outcomes of the second model [II] were defined in the following way: (a) a 90-day mRS score of 0, if the baseline NIHSS scores were < 8, (b) an mRS score of 0 or 1, if the NIHSS scores were 8–14, and (c) an mRS score of 0–2, if the NIHSS scores were > 14 points. The investigation involved 31 SCH patients and 725 thyroid-normal individuals. Similar clinical and laboratory characteristics existed between the SCH and control groups. Both groups had similar severity of stroke. 77% of SCH patients admitted had modest neurologic impairments, 6.5% had moderate impairments, and 16.1% had severe impairments. The distribution of initial neurologic impairments could be compared to that of the control group. Despite comparable initial deficits, the SCH group had a greater 30-day NIHSS score improvement rate (48.4% versus 25%). The 30-day outcomes of SCH patients (group [I]) were superior to those of group [II] patients (29.0%). At 90 days, the SCH group had superior [I] (74.2%) and [II] (58%) outcomes. Baek et al., discovered that patients with AIS may benefit from SCH. The functional outcomes at 90 days for ischaemic stroke patients with SCH at admittance were superior to those with normal thyroid function (58.1% vs. 31%). Both groups had equivalent severity scores on the NIHSS at admission, suggesting that undiagnosed SCH aided the recovery of stroke patients. In a prospective trial of first-time ischemic stroke patients, Akhoundi et al. [64] examined whether subclinical hypothyroidism (SCH) protects against stroke severity and prognosis using scales such as the NIHSS, mRS, and BI. Elevated serum TSH levels were associated with milder strokes upon admission, but this was not statistically significant. SCH patients exhibited significantly better functional outcomes than those with normal TSH levels after one month. Three months later, the results were similar. At the 1-month follow-up, 11 (22.5%) of the normal TSH group died, while the SCH group had no deaths. The elevated serum TSH group also showed lower 3-month mortality. A forward stepwise multiple linear regression analysis was performed to remove confounding variables from stroke outcomes. In the multivariate analysis, stroke severity was adversely related to positive outcomes, while serum TSH, TIA history, and WBC count were substantially associated. This study found that cerebral ischemic stroke patients with high blood TSH levels and no hypothyroidism had better functional outcomes at 1 and 3 months than those with normal TSH. They also found a decreased mortality rate in acute ischemic stroke patients between 1 and 3 months. They defined elevated blood TSH values as 2.5 mIU/L or higher based on the National Academy of Clinical Biochemistry's criteria for euthyroidism and preclinical hypothyroidism. Delpont et al. [65] examined TSH concentrations with clinical severity at admission and early functional prognosis in AIS patients. The NIHSS measured clinical severity upon admission, while the mRS assessed functional impairment at release. TSH levels were categorized by tertiles (<0.822, 0.822–1.6, >1.6 mUI/L). Depending on TSH

levels, 69.5% of the first tertile, 49.5% of the second, and 35.4% of the third experienced moderate-to-severe stroke (NIHSS score of 5 or more). In multivariable models, higher TSH levels at admission are linked to lower moderate-to-severe stroke rates. It was noted that when TSH concentrations increased, discharge mRS scores declined. Higher TSH levels at admission were linked to better functional outcomes when discharged, according to multivariable studies. Delpont et al., highlighted that a higher TSH level was independently linked to a lower severity score at admission and a better functional prognosis when patients with AIS were discharged.

### 4.3. Studies Concluding That There Is No Statistically Significant Evidence Connecting Thyroid Hormones with Stroke Outcomes

The classical "stress response" occurs when a stressor activates the hypothalamic-pituitary axis (HPA axis). Cortisol levels rise, thyroid function decreases, and growth hormone (GH) and insulin insufficiency develop. One of the initial alterations in brain ischemia is HPA-axis endocrine abnormalities. Neidert et al. [66] examined cortisol, T3, fT4, TSH, and GH on hospital admission to predict functional prognosis and mortality within 90 days and 1 year. Functional outcome was assessed using the mRS. The NIHSS correlated positively with cortisol and GH but negatively with T3 and TSH. NIHSS was unrelated to fT4 levels. Cortisol levels rose with lesion size in individuals with MRI data, while T3 and fT4 levels were unaltered. Patients with extensive lesions had lower TSH and higher GH levels. Patients with a good outcome and survivors had lower median cortisol levels, higher TSH and T3, lower fT4 levels, far greater T3/fT4 ratios, and lower GH concentrations than those with a bad outcome and non-survivors, respectively. Finally, univariate logistic regression models linked cortisol, T3, and TSH to functional results. Mortality was also linked to cortisol, fT4, and T3. Cortisol was the only pituitary axis hormone that independently predicted functional outcome and mortality in a logistic model adjusted for NIHSS and age. Cortisol was found to be an independent prognostic marker of functional outcome and death in ischemic stroke patients; however, it did not significantly contribute to the NIHSS clinical score. Cortisol levels increased with lesion size, neurological deficit (as measured by the NIHSS), and clinical stroke syndrome, indicating the severity of the stroke. In contrast, T3, fT4, TSH, and GH contribute barely any prognostic information to currently utilized measures and scores. O'Keefe et al. [67] studied thyroid hormone levels and ischemic stroke outcomes using the mRS and modified Barthel index (mBI) at hospital discharge and at 3 and 12 months. Before stroke, patients with low fT3 had higher mRS scores and lower mBI scores. The admission NIHSS was negatively linked with fT3, but not TSH or fT4. Hospital mRS was inversely linked to TSH and positively linked to fT4. There was no connection between fT3 and MRS. Additionally, fT3, fT4, and TSH levels did not affect admission mBI scores. Hospital mortality was increased for people with reduced TSH but not fT3 or fT4. However, discharged patients had higher fT3 levels than those who died in the hospital or went to hospice. Low TSH levels were linked to 3-month mortality. Patients who died had lower TSH levels at 3 months. Higher fT3 levels and mBI > 15 predicted a good 3-month prognosis. fT4 did not correlate with outcome, and individuals with a good composite outcome at 3 months had greater TSH levels than those with a bad composite outcome. The one-year outcome follow-up showed a similar pattern. As evaluated by the mBI, those with the highest fT3 levels one year after ischemic stroke were most independent. Even a year after the index event, higher initial fT3 levels were associated with better prognoses. At 12 months, neither the mBI nor the mRS were linked with TSH or fT4. Univariate analysis revealed potential confounding variables, which a multivariate logistic regression model controlled. Thyroid parameters were not associated with mortality or functional outcomes after controlling for AFIB, depression, UTI, IA, age, and NIHSS on admission, suggesting that thyroid state is linked to other risk factors. After adjusting for other stroke outcome indicators, this study indicated that lower free T3 levels were related to worse outcomes at hospital discharge and at 3 and 12 months post-stroke.

## 5. Study Limitations

Our study examines the potential association between thyroid function profile (thyroid hormone fluctuations within or outside normal limits) and stroke prognosis. We meticulously included all pertinent studies examining the relationship between thyroid profile and stroke prognosis in patients without overt thyroid disease. Our results should be interpreted with caution due to the large heterogeneity between studies in terms of study design and variation in cut-off values for thyroid hormone levels and primary endpoints. In addition, distinct scales and scores were used to assess stroke severity, recovery, and overall outcome in the aforementioned studies, making direct comparison difficult.

## 6. Conclusions

Emerging data regarding the potential predictive value of thyroid hormone levels suggests there may be a correlation between low T3 syndrome, subclinical hypothyroidism, and poor outcomes in stroke, particularly in certain age groups. Since low T3 levels may serve as a negative biomarker for stroke prognosis and aid in initiating early and intensive rehabilitation in these patients, this can be of vital importance in routine clinical practice. Measuring serum thyroid hormone concentrations is a non-invasive, relatively harmless, and safe screening test that may be useful for this purpose.

Low T3 serum levels are associated with worse functional outcomes, a higher incidence of post-stroke fatigue, higher NIHSS and mRS scores, a higher incidence of post-stroke infections (such as aspiration pneumonia and urinary tract infections), and a significantly poorer prognosis, according to research data. To securely conclude that there is a correlation between thyroid hormone levels and stroke prognosis, further research is necessary. In addition, prospective randomized trials are required to determine whether early hormone supplementation improves stroke outcomes for patients.

Importantly, there are substantial limitations to measuring T3 as a biomarker in acute settings. Low T3 syndrome is not unique to stroke patients; it is commonly observed in the critically ill. In addition, T3 levels should always be assessed in the context of the patient's medical history, as common thyroid diseases, thyroid hormone replacement therapy, nonthyroidal drugs such as amiodarone, and medications used for the treatment of acute stroke can affect the normal functioning of the HPT axis and alter T3 serum concentrations.

In general, subclinical hypothyroidism at admission appears to be associated with poorer functional outcomes following a stroke. However, these findings merit further research.

**Author Contributions:** A.G. and V.K. reviewed the literature, screened the abstracts of the reference list, deleted duplicates and citations not meeting the inclusion criteria, and assessed the articles; D.T. and K.V. resolved any disagreements regarding screening or the selection process; E.L., F.C., E.G. and S.K. wrote the first manuscript; C.K. (Christos Kokkotis), M.K., A.T., P.B., S.C. and C.K. (Christos Koutsokostas) reviewed the tables, the presentation of the data, and the methodology. The corrected version was discussed collegially. A.G., V.K., K.T., M.B., D.M. and N.A. wrote the final version. All authors have read and agreed to the published version of the manuscript.

**Funding:** This work was supported by the project "Study of the Interrelationships Between Neuroimaging, Neurophysiological, and Biomechanical Biomarkers in Stroke Rehabilitation (NEUROBIOMECH in Stroke Rehab)" (MIS 5047286), which is implemented under the action of "Support for Regional Excellence", funded by the operational program "Competitiveness, Entrepreneurship, and Innovation" (NSRFm2014-2020) and co-financed by Greece and the European Union (the European Regional Development Fund).

**Institutional Review Board Statement:** Not applicable.

**Informed Consent Statement:** Not applicable.

**Data Availability Statement:** All data discussed within this manuscript is available on PubMed.

**Conflicts of Interest:** The authors declare no conflict of interest.

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
