# Peer review of "Investigating the Predictive Value of Thyroid Hormone Levels for Stroke Prognosis"

_2035-8377, doi:10.3390/neurolint15030060_

Round 1
Reviewer 1 Report
Dear authors,
You have done a commendable job. The article looks fine to me in its present form and ready to get published.
Thanks,
The Reviewer

Author Response
Dear Reviewer,
Many thanks for your kind words and your time spent reviewing our manuscript.
Yours Sincerely
Dr Tsiptsios
Reviewer 2 Report
see below

minor revision
Author Response
Dear Reviewer,
Many thanks for your prompt response and the time spent reviewing our manuscript.
Your comments were thoroughly investigated and appropriate modifications in the text were made, as follows:
- First, we have divided Table 1 into three distinct tables based on their relationship to the outcome prognosis.
- Secondly, we have corrected the sub-numbering in Section 4.
- Finally, as for lines 334-335, we added a few lines considering the results of the studies you suggested.
Looking forward to your follow up comments.
Yours Sincerely
Dr Tsiptsios
Reviewer 3 Report
Dear authors, thank you for this comprehensive review.
Just one comment -please shorten the introduction. 50 - 60 lines would do. Especially lines 45-50 as well as sentence starting on line 33 -39 are not needed in such a paper as those are wellknown facts.
Author Response
Dear Reviewer,
Many thanks for your prompt response.
According to your suggestion we have shortened the lines 45-50 and the sentence starting on lines 33 -39. Consequently, the total length of the introduction was also diminished to some degree.
Looking forward to your follow up comments.
Yours Sincerely
Dr Tsiptsios
Round 2
Reviewer 2 Report
no comments